# Japanese clinical practice patterns of rituximab treatment for minimal change disease in adults 2021: A web-based questionnaire survey of certified nephrologists

Masahiro Koizumi[1], Takuji Ishimoto[2], Sayaka Shimizu[3,4,5]*, Sho Sasaki[6,7], Noriaki Kurita[5,8‡], Takehiko Wada[9‡]

1 Division of Nephrology, Endocrinology and Metabolism, Tokai University School of Medicine, Kanagawa, Japan, 2 Department of Nephrology and Rheumatology, Aichi Medical University, Aichi, Japan, 3 Section of Clinical Epidemiology, Department of Community Medicine, Kyoto University, Kyoto, Japan, 4 Patient Driven Academic League (PeDAL), Tokyo, Japan, 5 Department of Clinical Epidemiology, Graduate School of Medicine, Fukushima Medical University, Fukushima, Japan, 6 Section of Education for Clinical Research, Kyoto University Hospital, Kyoto, Japan, 7 Center for Innovative Research for Communities and Clinical Excellence, Fukushima Medical University, Fukushima, Japan, 8 Department of Innovative Research and Education for Clinicians and Trainees (DiRECT), Fukushima Medical University Hospital, Fukushima, Japan, 9 Department of Nephrology, Toranomon Hospital, Tokyo, Japan

☯ These authors contributed equally to this work.
‡ NK and TW also contributed equally to this work.
* ssayaka.tkshm@gmail.com

**Data Availability Statement:** The minimal data can be found within the article and its accompanying Supporting Information files. However, as the

## Abstract

### Background

In Japan, rituximab (RTX) for adult-onset frequently relapsing (FR)/steroid-dependent (SD) minimal change disease (MCD) is not explicitly reimbursed by insurance, and its standard regimen has not been established.

### Methods

We conducted a cross-sectional web-based survey between November and December 2021. The participants were nephrologists certified by the Japanese Society of Nephrology and answered 7 items about RTX for adult MCD. Factors related to the experience of RTX administration at their facilities were estimated by generalized estimating equations.

### Results

Of 380 respondents, 181 (47.6%) reported the experience of RTX use for adult MCD at their current facilities. Those who worked at university hospitals (vs. non-university hospitals, proportion difference 13.7%) and at facilities with frequent kidney biopsies (vs. 0 cases/year, 19.2% for 1–40 cases/year; 37.9% for 41–80 cases/year; 51.9% for $\geq$ 81 cases/year) used RTX more frequently. Of 181 respondents, 28 (15.5%) answered that there was no insurance coverage for RTX treatment. Of 327 respondents who had the opportunity to treat

survey results may contain potentially identifiable participant information, they cannot be openly shared. For inquiries concerning access to this data, please reach out to the administrative office of General Incorporated Association PeDAL at admin@pedal.or.jp.

**Funding:** This study was partly supported by a Grant-in-Aid for Intractable Renal Diseases Research, Research on Rare and Intractable Diseases, and Health and Labor Sciences Research Grants from the Ministry of Health, Labour and Welfare of Japan (ID: 20FC1045). The funders had no role in study design, data collection and analysis, decision to publish, or preparation of the manuscript.

**Competing interests:** We have read the journal's policy and the authors of this manuscript have the following competing interests: Noriaki Kurita (Honoraria from GlaxoSmithKline). This does not alter our adherence to PLOS ONE policies on sharing data and materials.

MCD, which was a possible indication for RTX, 178 (54.4%) indicated withholding of RTX administration. The most common reason was the cost due to lack of insurance coverage (141, 79.2%). Regarding RTX regimens for FR/SD MCD, introduction treatment with a single body surface area-based dose of 375 mg/m$^2$ and maintenance treatment with a 6-month interval were the most common.

## Conclusion

This survey revealed the nephrologists' characteristics associated with RTX use, the barriers to RTX use, and the variation in the regimens for adult MCD in Japan.

## Introduction

The medical treatment of minimal change disease (MCD) with frequent relapses and steroid dependency is a major challenge in clinical practice. As a result, achieving adequate control of MCD remains difficult even when combined therapy with corticosteroids and immunosuppressive agents is used [1]. Rituximab (RTX) is one of the currently accepted drugs as an effective treatment for frequently relapsing (FR)/steroid-dependent (SD) MCD. RTX has been successfully verified for its efficacy and safety in a randomized controlled trial (RCT) for childhood-onset FR/SD MCD [2] and has been positioned as one of the treatment options for FR/SD nephrotic syndrome in the latest Kidney Disease: Improving Global Outcomes (KDIGO) guidelines [3]. In the Japanese guidelines, RTX is listed in the treatment algorithm for MCD in adults [4, 5]. Nevertheless, the application of RTX in adult-onset FR/SD MCD in Japan remains uncertain due to the lack of supporting evidence for its efficacy and its exclusion from explicit medical reimbursement.

RTX for FR/SD MCD in adults was suggested to induce complete remission in more than 90% of cases and subsequently prevent relapse by a meta-analysis of several small observational studies [6]. However, the efficacy of RTX has not been established because no RCTs have been conducted for adult-onset cases. In addition, reimbursement of RTX in Japan is only approved for childhood-onset FR/SD MCD. Clarifying the current Japanese experience with RTX and the existing barriers to its use in adult cases may be useful in developing a treatment protocol and generating evidence for insurance reimbursement of RTX therapy for adult-onset FR/SD MCD.

Therefore, we conducted a web-based survey of Japanese nephrologists to analyze RTX treatment patterns and the reasons for barriers to its use for MCD in adults.

## Materials and methods

### Study design and setting

This was a cross-sectional web-based survey using Microsoft Forms (Microsoft, Redmond, WA, USA) conducted between November 15 and December 31, 2021. Detailed methods are summarized elsewhere [7]. This study was an anonymous survey of healthcare professionals to describe practices regarding primary nephrotic syndrome and was considered outside the scope of ethical review according to the Ethical Guidelines for Medical and Biological Research Involving Human Subjects [8]. The possibility of academic publication of the survey results was described at the beginning of the questionnaire, and only those who provided consent to complete the survey were included.

## Participants

The target population was nephrologists certified by the Japanese Society of Nephrology (JSN). There were 5777 certified nephrologists at the time of the survey [9]. The sampling method used was convenience sampling, using the mailing list for JSN members or direct mailing by members of the working group to nephrologists of their acquaintance [7].We excluded respondents who were not currently involved in caring for patients with primary nephrotic syndrome in an outpatient setting and respondents who did not provide the identifiable zip code of their affiliation, based on their responses to relevant items in the survey form. The number of clinics (defined as medical facilities having ≤ 20 inpatient beds) and hospitals offering nephrology as a medical specialty in 2020 is 2154 and 1381, respectively [10]. As of July 18, 2023, there are 715 teaching facilities accredited by the JSN [11]. Although the exact number of facilities performing kidney biopsies is unavailable, we believe that the number is close to the number of these accredited teaching facilities.

## Data collection methods

The survey consisted of 34 questionnaire items, and 7 items were related to RTX treatment for MCD with/without frequent relapses in adults. All response options were multiple choice, with open-ended responses allowed for part of the items. Each question was asked only for participants who currently had the opportunity to treat patients with the relevant disease with RTX. Details of the 7 question items are shown in S1 Text. Briefly stated, these seven items asked about 1) the experience of RTX administration for adult MCD at the current affiliation, 2) the funding source of RTX treatment for adult MCD at the current affiliation, 3) the intention to withhold RTX for adult MCD that was supposedly to be responsive to RTX, and 4) the RTX regimen for FR cases of MCD (dose, intervals, and frequency during introduction/maintenance period, duration of the maintenance period). Information on participants' backgrounds, years of experience as a physician, type of affiliation, number of kidney biopsies performed at the affiliation, number of patients with primary nephrotic syndrome treated in an outpatient setting, and zip codes of the affiliation was collected. According to the zip codes, the region and the population density of the affiliation location were identified.

## Statistical analyses

Categorical variables are expressed as numbers and percentages. As all items were required to be answered, there were no missing values. Open-ended responses were reviewed by at least two nephrologists and classified into either existing choice categories or newly created categories. A comparison between groups with and without the experience of RTX use at the current affiliation (the RTX administration group and the non-administration group, respectively) was performed by Pearson's chi-square test. To explore factors associated with the experience of RTX use at the current affiliation, a generalized estimating equation with robust variance estimation (with an identity link function and a Gaussian distribution family) was fitted with the following independent variables: years of experience as a physician, type of affiliation, number of kidney biopsies performed at the affiliation, number of patients with primary nephrotic syndrome treated in an outpatient setting, and population density of the location of the affiliation [12]. The proportion difference in usage experience was estimated as an effect measure. Sankey diagrams are presented to provide a graphical overview of the distribution of choices for the introduction and maintenance treatment protocol of FR/SD MCD in adults [13]. All analyses were performed using STATA version 17 (StataCorp LLC, College Station, TX, USA).

## Results

Of the overall 434 respondents, 48 were not currently engaged in medical treatment of patients with primary nephrotic syndrome in an outpatient setting, and the zip codes of 6 respondents were not identified. As a result, the data of 380 respondents from 278 facilities were analyzed in this study. The locations of which the ZIP codes (assuming there was only one nephrology provider in that district) were correctly identified among the 278 facilities are shown in Fig 1. The characteristics of these 380 respondents are summarized in Table 1.

### 1. Characteristics of nephrologists with and without the experience of RTX use for adult MCD and its funding sources at the current facility

Among the respondents, 181 (47.6%) reported the experience of RTX use for adult MCD. Specifically, 33 (8.7%), 115 (30.3%), 24 (6.3%), and nine (2.4%) answered that they had an average number of less than one, 1 to 5, 6 to 20, and over 20 such cases per year, respectively. A total of 199 respondents (52.4%) had no experience with RTX use, and 35 (9.2%) referred their patients to other facilities where RTX was available. The participants' characteristics grouped by the experience of RTX administration are summarized in Table 2.

From the multivariable analysis (Table 3), the respondents who worked at university hospitals used RTX more frequently than those at non-university hospitals (proportion difference

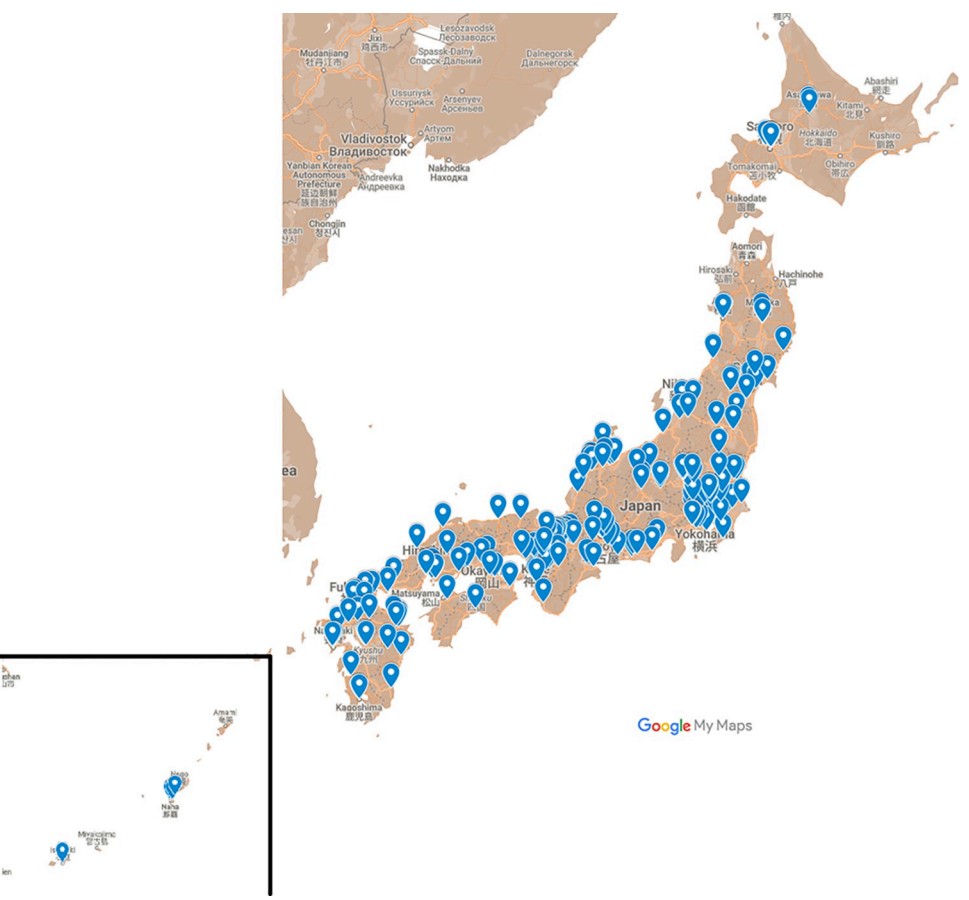

**Fig 1. Nationwide map of the responding nephrologists' workplace.** Of the 380 respondents, 278 unique, identifiable zip codes for their workplaces were provided, and 268 of which are mapped onto Google My Maps. To view the actual distribution via Google My Maps, please click on the following link: http://tinyurl.com/4j2ct6s9.

**Table 1. Respondent characteristics (n = 380).**

|  | n | (%) |
|---|---|---|
| Experience |  |  |
| ≤ 10 years | 42 | (11.1%) |
| 11 to 20 years | 151 | (39.7%) |
| 21 to 30 years | 125 | (32.9%) |
| 31 years ≤ | 62 | (16.3%) |
| Affiliation |  |  |
| General hospital | 201 | (52.9%) |
| University hospital | 153 | (40.3%) |
| Clinic | 26 | (6.8%) |
| Number of kidney biopsies performed at the facility (per year) |  |  |
| None | 55 | (14.5%) |
| ≤ 40 | 118 | (31.1%) |
| 41 to 80 | 115 | (30.3%) |
| 81 ≤ | 92 | (24.2%) |
| Number of patients with primary nephrotic syndrome treated in an outpatient setting per participant (per month) |  |  |
| 1 to 4 | 106 | (27.9%) |
| 5 to 14 | 181 | (47.6%) |
| 15 ≤ | 93 | (24.5%) |
| Location of affiliation |  |  |
| Hokkaido | 8 | (2.1%) |
| Tohoku | 24 | (6.3%) |
| Kanto | 127 | (33.4%) |
| Chubu | 60 | (15.8%) |
| Kinki | 73 | (19.2%) |
| Chugoku | 37 | (9.7%) |
| Shikoku | 8 | (2.1%) |
| Kyushu/Okinawa | 43 | (11.3%) |

13.7%, 95% confidence interval [CI] 1.7–25.7%, p = 0.03). Those who belonged to a facility where more kidney biopsies were performed also used RTX more frequently (with 0 cases/year as the reference, 19.2% [95% CI 3.9–34.5%, p = 0.01] for 1–40 cases/year; 37.9% [21.2–54.6%, p < 0.01] for 41–80 cases/year; 51.9% [33.9–69.9%, p < 0.01] for ≥ 81 cases/year).

Among 181 who had experience with RTX use, the funding source for RTX expenses was as follows: 140 (77.3%) reported that the costs were paid by patients with coverage by medical insurance, 24 (13.3%) reported that the costs were covered by patient self-payment or hospital payment without coverage by insurance, and 4 (2.2%) reported that the costs were covered by research funding from the clinical departments (Table 4).

## 2. Withholding of RTX administration for MCD in adults

Of 327 respondents in charge of adults with MCD that was supposedly responsive to RTX treatment, 178 (54.4%) indicated that they either had withheld or would withhold RTX administration. The most common reason was the inability to afford the financial costs due to lack of medical insurance coverage (141, 79.3%), followed by limited experience in the usage of RTX (31, 17.4%) (Table 5).

**Table 2. Characteristics of respondents with and without experience of rituximab use for adult minimal change disease in the current facility (n = 380).**

| | Administration group | Non-administration group | p value |
|---|---|---|---|
| | (n = 181) | (n = 199) | |
| Experience | | | 0.25 |
| ≤ 10 years | 19 (10.5%) | 23 (11.6%) | |
| 11 to 20 years | 81 (44.8%) | 70 (35.2%) | |
| 21 to 30 years | 52 (28.7%) | 73 (36.7%) | |
| 31 years ≤ | 29 (16.0%) | 33 (16.6%) | |
| Affiliation | | | < 0.001 |
| Non-university hospital | 77 (42.5%) | 124 (62.3%) | |
| University hospital | 102 (56.4%) | 51 (25.6%) | |
| Clinic | 2 (1.1%) | 24 (12.1%) | |
| Number of kidney biopsies performed at the facility (per year) | | | < 0.001 |
| None | 4 (2.2%) | 51 (25.6%) | |
| ≤ 40 | 38 (21.0%) | 80 (40.2%) | |
| 41 to 80 | 70 (38.7%) | 45 (22.6%) | |
| 81 ≤ | 69 (38.1%) | 23 (11.6%) | |
| Number of patients with primary nephrotic syndrome treated in an outpatient setting per participant (per month) | | | < 0.001 |
| 1 to 4 | 27 (14.9%) | 79 (39.7%) | |
| 5 to 14 | 92 (50.8%) | 89 (44.7%) | |
| 15 ≤ | 62 (34.3%) | 31 (15.6%) | |
| Location of affiliation | | | < 0.001 |
| Hokkaido | 5 (2.8%) | 3 (1.5%) | |
| Tohoku | 19 (10.5%) | 5 (2.5%) | |
| Kanto | 71 (39.2%) | 56 (28.1%) | |
| Chubu | 34 (18.8%) | 26 (13.1%) | |
| Kinki | 29 (16.0%) | 44 (22.1%) | |
| Chugoku | 5 (2.8%) | 32 (16.1%) | |
| Shikoku | 3 (1.7%) | 5 (2.5%) | |
| Kyushu/Okinawa | 15 (8.3%) | 28 (14.1%) | |
| Population density of the location of the affiliation (per square kilometer) | | | < 0.001 |
| ≤ 999 | 52 (28.7%) | 85 (42.7%) | |
| 1000 to 5000 | 55 (30.4%) | 55 (27.6%) | |
| 5001 to 9999 | 20 (11.0%) | 30 (15.1%) | |
| 10000 ≤ | 54 (29.8%) | 29 (14.6%) | |

## 3. Regimens of RTX for FR nephrotic syndrome

Of 380 respondents, 124 were excluded because they had limited experience in using RTX and could not answer the questions about the regimens; the data from the remaining 256 respondents were analyzed.

The relationship between the dose and the treatment protocol during the introduction period is illustrated in Fig 2. During the introduction period, the majority of respondents used a body surface area (BSA)-based dosage of 375 mg/m$^2$ with an upper threshold of 500 mg (213, 83.2%). Regarding the administration protocols, 155 (60.5%) responded that they used

**Table 3. Analysis of factors associated with the usage experience of rituximab in the current facility (n = 380).**

| | Proportion difference, point estimate | 95% confidence interval | p value |
|---|---|---|---|
| Experience | | | |
| ≤ 10 years | Reference | | |
| 11 to 20 years | 0.065 | (-0.066, 0.197) | 0.33 |
| 21 to 30 years | -0.078 | (-0.220, 0.064) | 0.28 |
| 31 years ≤ | 0.051 | (-0.107, 0.209) | 0.53 |
| Affiliation | | | |
| Non-university hospital | Reference | | |
| University hospital | 0.137 | (0.017, 0.257) | 0.03 |
| Clinic | 0.017 | (-0.153, 0.187) | 0.85 |
| Number of kidney biopsies performed at the facility (per year) | | | |
| None | Reference | | |
| ≤ 40 | 0.192 | (0.039, 0.345) | 0.01 |
| 41 to 80 | 0.379 | (0.212, 0.546) | < 0.01 |
| 81 ≤ | 0.519 | (0.339, 0.699) | < 0.01 |
| Number of patients with primary nephrotic syndrome treated in an outpatient setting per participant (per month) | | | |
| 1 to 4 | Reference | | |
| 5 to 14 | 0.093 | (-0.001, 0.187) | 0.05 |
| 15 ≤ | 0.136 | (-0.0003, 0.273) | 0.05 |
| Population density of the location of the affiliation (per square kilometer) | | | |
| ≤ 999 | Reference | | |
| 1000 to 5000 | 0.007 | (-0.113, 0.127) | 0.91 |
| 5001 to 9999 | -0.088 | (-0.249, 0.073) | 0.29 |
| 10000 ≤ | 0.105 | (-0.030, 0.241) | 0.13 |

A generalized estimating equation was fit to estimate the proportionate difference in the experience while considering the clustering of nephrologists in the same facility with the independent variables of years of experience as a physician, type of affiliation, number of kidney biopsies performed at the affiliation, number of patients with primary nephrotic syndrome treated in an outpatient setting, and the population density of the location of the affiliation.

single-dose administration, followed by four times at one-week intervals (45, 17.6%) and twice at one-week intervals (36, 14.1%). Of the 213 respondents who used a BSA-based dose of 375 mg/m$^2$, 125 used single-dose administration, followed by four times at one-week intervals (42 respondents) and twice at one-week intervals (30 respondents).

The relationship between the treatment interval and the treatment duration is illustrated in Fig 3. During the maintenance period with sustained remission, 158 respondents (61.7%) indicated that the interval of RTX administration was 6 months, and 63 (24.6%) had no plan for

**Table 4. Funding sources of rituximab treatment for adult minimal change disease cases (n = 181).**

| | n | (%) |
|---|---|---|
| Patient payment (covered by insurance) | 140 | (77.3%) |
| Unknown | 16 | (8.8%) |
| Hospital payment (not covered by insurance) | 15 | (8.3%) |
| Patient payment (not covered by insurance) | 9 | (5.0%) |
| Research funding from the clinical department | 4 | (2.2%) |
| Research funding from an individual doctor | 0 | (0%) |
| Others | 4 | (2.2%) |

**Table 5. Reasons for withholding rituximab for minimal change disease in adults (n = 178).**

| | n | (%) |
|---|---|---|
| Inability to afford financial costs due to lack of medical insurance coverage | 141 | (79.3%) |
| Limited experience in its usage | 31 | (17.4%) |
| Prohibition of its usage by the facility or the ethical committee due to lack of medical insurance coverage | 19 | (10.7%) |
| Inadequate medical care system in the facility for possible complications of RTX treatment | 12 | (6.7%) |
| Others | 5 | (2.8%) |

RTX treatment. Four respondents (1.6%, categorized into the "others" category in Fig 2) reported administration based on the CD19/20-positive cell count. The most common total duration of treatment was 1 to 2 years (55, 21.5%), followed by 2 to 3 years and more than 3 years (51, 19.9%, both). Three respondents (1.2%, categorized into the "others" category in Fig 2) determined the duration of treatment based on the CD19/20-positive cell count. The treatment duration had high variability, independent of the treatment interval.

## Discussion

In this web-based questionnaire study among nephrologists, we reported the actual situation of RTX treatment for MCD in adults in Japan. Approximately half of the respondents had experience with RTX use for adult MCD at their current affiliation, and many of them belonged to university hospitals or medical institutions where kidney biopsies were commonly performed. While medical insurance covered the majority of RTX treatment expenses, 15.5% of the respondents reported that the cost was not paid by insurance. Moreover, more than half

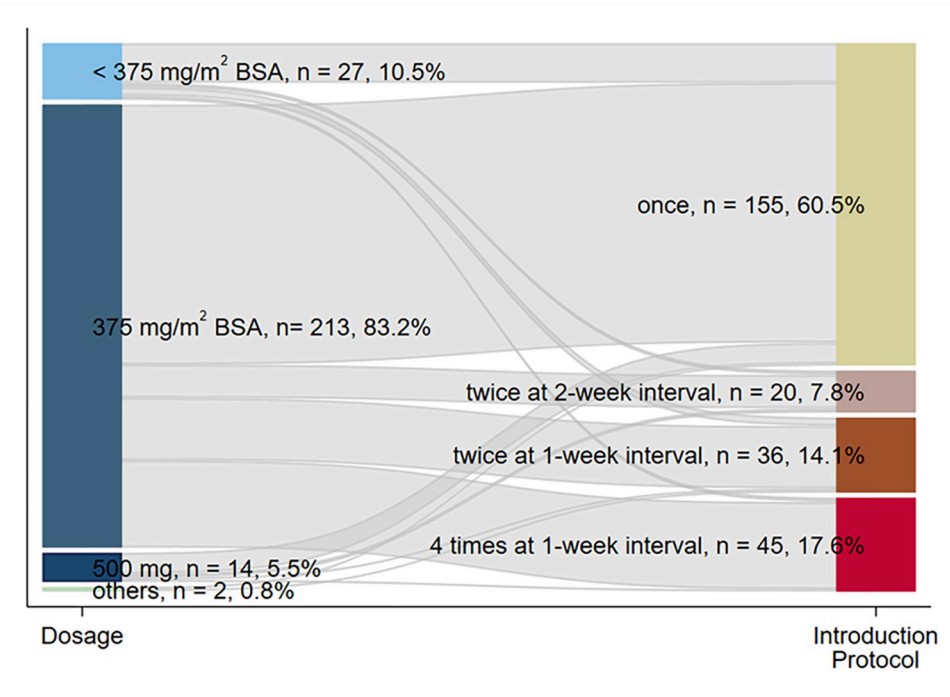

**Fig 2. The distribution and combination of the rituximab treatment protocol in the introduction period presented by a Sankey diagram.** BSA, body surface area.

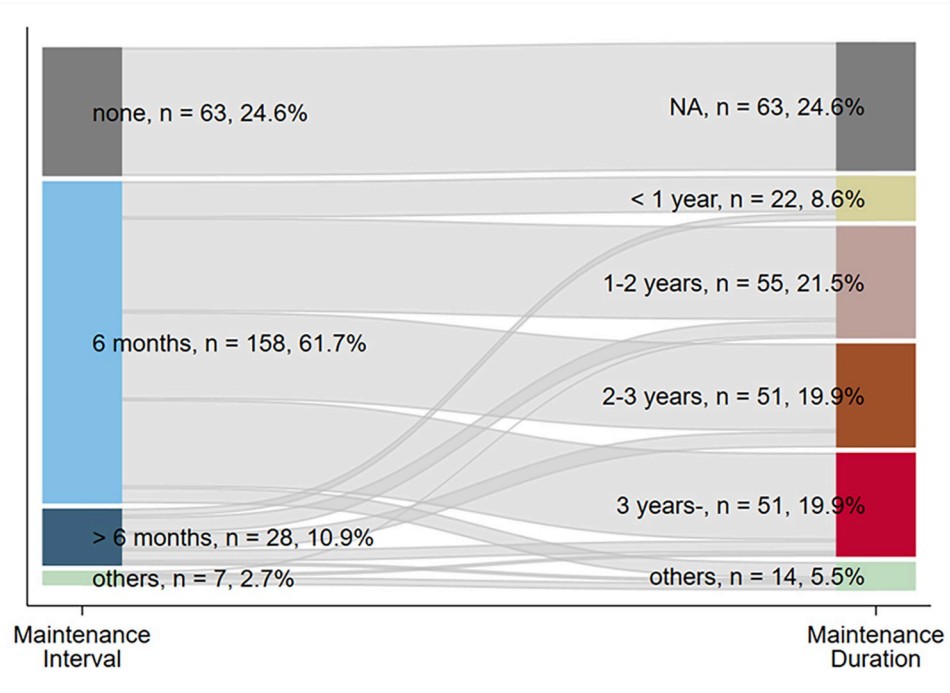

**Fig 3. The distribution and combination of the rituximab treatment protocol in the maintenance period presented by a Sankey diagram.** NA, not applicable.

of the respondents had either withheld or would withhold RTX treatment for MCD in adults, mainly due to the inability to afford the financial cost. As for RTX for FR/SD MCD in adults, although there was substantial variability among the treatment protocols, the most common response was introduction treatment with a single BSA-based dose of 375 mg/m$^2$ and maintenance treatment with a 6-month interval.

Several factors may account for the significantly higher RTX utilization among nephrologists affiliated with university hospitals or facilities with a high frequency of kidney biopsies. First, such facilities tend to have specialized medical departments and well-organized treatment provision systems, making access to RTX easier. As such, a protocol for managing infusion reactions that may occur during RTX treatment is also well established. Second, nephrologists working at such facilities are more likely to encounter refractory MCD cases for which RTX is a good treatment indication and, as a result, are more likely to gain experience in RTX use.

The present findings of a nonnegligible percentage of full payment for RTX therapy by institutions or patients and over half of the respondents withholding RTX due to the inability to afford RTX expenses underscore the need for insurance reimbursement for FR/SD MCD in adults. Indeed, an excellent cost-effectiveness of RTX therapy has been reported. RTX therapy with a total of four doses every 6 months with 500 mg has been shown to not only reduce relapse but also cut 56% of medical costs [14]. RTX therapy based on CD19 monitoring has also been shown to have superior benefits in medical costs [15]. On the other hand, this study also revealed that the majority of RTX use in adult FR/SD MCD cases was approved for reimbursement without being assessed as inappropriate by the authorities.

In introduction therapy for FR/SD MCD in adults, a BSA-based dose of 375 mg/m$^2$ was most frequently employed, but its frequency varied, with a single dose being the most

common. This Japanese practice pattern differed from the four weekly administrations indicated on the RTX label. First, the four weekly doses indicated on the label were merely extrapolated from the dosing for malignant lymphoma [16]. Second, although studies have reported the effectiveness of this dosing protocol of RTX for MCD in adults [17, 18], there has been growing interest in treatment protocols with less frequent dosing. Indeed, treatment outcomes with BSA-based doses of 375 mg/m$^2$ and single or double doses have been reported for childhood- and adult-onset FR/SD MCD [19–21]. On the other hand, it is noteworthy that several respondents chose a substandard dosage. This reality may reflect findings from both Japanese and foreign studies that reported the effectiveness of a single 200-mg dose of RTX [22, 23].

Regarding maintenance therapy, the large proportion of choices for regular dosing at 6-month intervals may reflect those intervals employed in clinical trials conducted in Japan [20, 21, 24]. In contrast, a certain proportion of the respondents reported no periodic dosing. This may reflect nephrologists' concerns about the increased incidence of adverse events such as infections, the production of anti-chimeric antibodies to RTX, and hypogammaglobulinemia reported in the pediatric setting [25]. Alternatives to periodic RTX dosing include mycophenolate mofetil [26] and RTX administration based on CD19-positive cell monitoring [27]. In this study, a minority of the respondents opted for the latter option. For the identification of the optimal treatment strategy for adult FR/SD MCD, further studies with patient data to analyze the effectiveness of those RTX regimens and doses are warranted. If patient-reported outcomes could be added to such studies, differences in treatment adherence, satisfaction, and quality of life would also be revealed.

The strength of this study was our enrollment of nephrologists engaged in the management of nephrotic syndrome nationwide to describe the real-world practice of RTX therapy for MCD in adults. In addition, the variation in introduction protocols and the intervals and durations of the maintenance period were straightforwardly demonstrated by Sankey diagrams. At the same time, there are several limitations. First, web-based convenience sampling may have introduced selection bias, and the responses in this study may not accurately reflect actual nationwide patterns. In addition, the total number of individuals who were invited to complete the survey was unavailable. Second, because the survey was anonymous, individuals may have provided more than one response. However, the dedication required for participation makes multiple responses unlikely. Third, there may have been information bias due to self-reporting on the percentages of RTX use, such as recall bias and social desirability bias. To determine a more accurate actual status of RTX use, a large database study linking patient data from medical records with physician data is necessary. Finally, we were not able to examine the regulations regarding RTX use on a facility basis. However, we believe that physician-level summary data would also be useful, for example, the funding sources of RTX treatment in Table 4, because discretionary authority may vary by job position even among nephrologists at the same facility.

## Conclusion

This nationwide survey of Japanese nephrologists on RTX practice patterns for MCD in adults revealed the characteristics associated with the treatment propensity, variation in the protocol during the introduction and maintenance periods, and the actual burden of the treatment costs and withholding of RTX in adult MCD cases. Currently, an RCT of RTX for MCD in adults (A-TEAM study) is underway in Japan, and the efficacy of a protocol of two BSA-based doses of 375 mg/m$^2$ weekly during the introduction phase, followed by a single dose during the maintenance phase at 6 months after the introduction, will be clarified [28]. Together with future results from the clinical trial, the present findings are expected to contribute to the establishment of a standard of care with RTX for FR/SD MCD in adults.

## Supporting information

**S1 Text. Questionnaire items.**
(DOCX)

## Acknowledgments

We would like to thank Springer Nature Author Services for English language editing.

## Author Contributions

**Conceptualization:** Takuji Ishimoto, Sayaka Shimizu, Noriaki Kurita, Takehiko Wada.

**Data curation:** Sayaka Shimizu, Noriaki Kurita, Takehiko Wada.

**Formal analysis:** Sayaka Shimizu, Sho Sasaki, Noriaki Kurita.

**Writing – original draft:** Masahiro Koizumi.

**Writing – review & editing:** Masahiro Koizumi, Takuji Ishimoto, Sayaka Shimizu, Sho Sasaki, Noriaki Kurita, Takehiko Wada.

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
