## [Decision Letter · Decision Letter 0]

16 Nov 2023

PONE-D-23-34256Japanese Clinical Practice Patterns of Rituximab Treatment for Minimal Change Disease in Adults 2021: A Web-Based Questionnaire Survey of Certified NephrologistsPLOS ONE

Dear Dr. Shimizu,

Thank you for submitting your manuscript to PLOS ONE. After careful consideration, we feel that it has merit but does not fully meet PLOS ONE’s publication criteria as it currently stands. Therefore, we invite you to submit a revised version of the manuscript that addresses the points raised during the review process.

We look forward to receiving your revised manuscript.

Kind regards,

Rajendra Bhimma, PhD

Academic Editor

PLOS ONE

Journal Requirements:

"This study was partly supported by a Grant-in-Aid for Intractable Renal Diseases Research, Research on Rare and Intractable Diseases, and Health and Labor Sciences Research Grants from the Ministry of Health, Labour and Welfare of Japan (ID: 20FC1045). The funder had no role in this study's design, conduct, or reporting"

6. Thank you for stating the following in the Competing Interests section: 

"I have read the journal's policy and the authors of this manuscript have the following competing interests: Noriaki Kurita (Honoraria from GlaxoSmithKline)"

Additional Editor Comments :

Please see comments by both reviewers and address each of these.

Reviewers' comments:

Reviewer's Responses to Questions

**Comments to the Author**

1. Is the manuscript technically sound, and do the data support the conclusions?

Reviewer #1: No

Reviewer #2: Partly

2. Has the statistical analysis been performed appropriately and rigorously? 

Reviewer #1: Yes

Reviewer #2: Yes

3. Have the authors made all data underlying the findings in their manuscript fully available?

Reviewer #1: Yes

Reviewer #2: Yes

4. Is the manuscript presented in an intelligible fashion and written in standard English?

Reviewer #1: Yes

Reviewer #2: No

5. Review Comments to the Author

Reviewer #1: General comments

This manuscript described the Japanese clinical practice patterns of rituximab treatment for minimal change disease in adults from nation-wide, cross-sectional questionnaire survey. Very important issue, but there are several concerns that should be addressed.

Comments

1. Table 3: Please specify the confounding factors that were entered into the multivariate analysis; was location of affiliation not included?

2. The survey results are shown for each individual, but are the results the same for each facility?

3. Why is "Patient payment (covered by insurance)" done even though it is not provided by insurance coverage?

Reviewer #2: This article investigated the clinical practice patterns of rituximab treatment for minimal change disease (MCD) in adults in Japan. The authors conducted a web-based survey of certified nephrologists and analyzed the factors associated with the use of rituximab, the barriers to its use, and the variation in the treatment regimens for adult MCD. It addresses a relevant and timely topic, as rituximab is a promising therapy for frequently relapsing or steroid-dependent MCD, but its efficacy, safety, and cost-effectiveness in adults are not well established, and its reimbursement by insurance is not explicit in Japan.

Weaknesses of this article also include:

• It relies on self-reported data, which may be subject to recall bias, social desirability bias, or inaccurate reporting by the respondents.

• It does not compare the outcomes or effectiveness of different rituximab regimens or dosages, which limits the ability to draw conclusions about the optimal treatment strategy for adult MCD.

• It does not explore the patients’ perspectives, preferences, or experiences with rituximab treatment, which may affect the adherence, satisfaction, and quality of life of the patients.

• It does not discuss the limitations, implications, or recommendations of the study, which may reduce the impact and applicability of the findings.

6. PLOS authors have the option to publish the peer review history of their article (what does this mean?). If published, this will include your full peer review and any attached files.

Reviewer #1: No

Reviewer #2: No

---

## [Author Response · Author response to Decision Letter 0]

1 Dec 2023

December 2, 2023

Emily Chenette

Editor-in-Chief, PLOS ONE

Dear Editor:

We wish to resubmit our manuscript titled “Japanese Clinical Practice Patterns of Rituximab Treatment for Minimal Change Disease in Adults 2021: A Web-Based Questionnaire Survey of Certified Nephrologists.” We would like to thank the reviewers and editors for their time and effort in reviewing this manuscript. Please consider the attached manuscript which has been revised according to the reviews. Please refer to the attached response letter. In the revised manuscript, edits made based on Reviewer 1's comments are highlighted in green, and Reviewer 2's, in yellow.

Regarding funding information, this study was partly supported by a Grant-in-Aid for Intractable Renal Diseases Research, Research on Rare and Intractable Diseases, and Health and Labor Sciences Research Grants from the Ministry of Health, Labour and Welfare of Japan (ID: 20FC1045). The funders had no role in study design, data collection and analysis, decision to publish, or preparation of the manuscript. Regarding competing interests, we have read the journal's policy and the authors of this manuscript have the following competing interests: Noriaki Kurita (Honoraria from GlaxoSmithKline). This does not alter our adherence to PLOS ONE policies on sharing data and materials. We have removed the relevant text from the manuscript and would appreciate it if the journal office could change the description in the online submission form. English language editing was performed by Springer Nature Author Services.

Thank you for considering our manuscript. We are honored to have our manuscript published in PLOS ONE journal.

Sincerely,

Corresponding author

Sayaka Shimizu, MD, PhD

Section of Clinical Epidemiology, Department of Community Medicine, Graduate School

of Medicine, Kyoto University, Kyoto, Japan

Tel: +81-75-366-7655, E-mail: ssayaka.tkshm@gmail.com

---

## [Decision Letter · Decision Letter 1]

18 Dec 2023

PONE-D-23-34256R1Japanese Clinical Practice Patterns of Rituximab Treatment for Minimal Change Disease in Adults 2021: A Web-Based Questionnaire Survey of Certified NephrologistsPLOS ONE

Dear Dr. Shimizu,

Thank you for submitting your manuscript to PLOS ONE. After careful consideration, we feel that it has merit but does not fully meet PLOS ONE’s publication criteria as it currently stands. Therefore, we invite you to submit a revised version of the manuscript that addresses the points raised during the review process.

**ACADEMIC EDITOR:**The manuscript is of interest to the nephrology community, but requires more detailed methodology description:

How many nephology in-patient and out-patient centers are there in Japan?

How many of these centers perform kidney biopsies?                                                                                                          

How many nephrologists certified by the Japanese Society of Nephrology (JSN) were practicing in Japan at the time of the survey? How may were invited to participate in the survey? How many responded to the invitation?

Please provide a map of the centers, indicating the number of practicing nephrologists and the number of nephrologists who completed the survey.

We look forward to receiving your revised manuscript.

Kind regards,

Justyna Gołębiewska

Academic Editor

PLOS ONE

Journal Requirements:

Reviewers' comments:

Reviewer's Responses to Questions

**Comments to the Author**

1. If the authors have adequately addressed your comments raised in a previous round of review and you feel that this manuscript is now acceptable for publication, you may indicate that here to bypass the “Comments to the Author” section, enter your conflict of interest statement in the “Confidential to Editor” section, and submit your "Accept" recommendation.

Reviewer #1: All comments have been addressed

Reviewer #2: (No Response)

2. Is the manuscript technically sound, and do the data support the conclusions?

Reviewer #1: Yes

Reviewer #2: No

3. Has the statistical analysis been performed appropriately and rigorously? 

Reviewer #1: Yes

Reviewer #2: N/A

4. Have the authors made all data underlying the findings in their manuscript fully available?

Reviewer #1: Yes

Reviewer #2: No

5. Is the manuscript presented in an intelligible fashion and written in standard English?

Reviewer #1: Yes

Reviewer #2: Yes

6. Review Comments to the Author

Reviewer #1: The authors responded appropriately to the revision of the paper. The content has been improved and is useful information for readers. Thank you for your efforts.

Reviewer #2: (No Response)

7. PLOS authors have the option to publish the peer review history of their article (what does this mean?). If published, this will include your full peer review and any attached files.

Reviewer #1: No

Reviewer #2: No

---

## [Author Response · Author response to Decision Letter 1]

12 Jan 2024

Detailed Response to Reviewers

Title: Japanese Clinical Practice Patterns of Rituximab Treatment for Minimal Change Disease in Adults 2021: A Web-Based Questionnaire Survey of Certified Nephrologists

We sincerely appreciate all the editor’s valuable comments. They helped us improve our manuscript, and we have taken them into account while revising our manuscript, as described below. Our revisions in the manuscript made based on the editor's comments are highlighted in yellow.

Academic editor’s comments:

The manuscript is of interest to the nephrology community, but requires more detailed methodology description.

1. How many nephology in-patient and out-patient centers are there in Japan?

Response: The number of clinics (defined as medical facilities having ≤20 inpatient beds) and hospitals offering nephrology as a medical specialty in 2020 is 2154 and 1381, respectively [1]. We have added this to the Participants portion of the Methods section.

Reference

1. Ministry of Health, Labour and Welfare. Summary of Static/Dynamic Survey of Medical Institutions and Hospital Report, 2020 (in Japanese). In: Ministry of Health, Labour and Welfare [Internet]. 27 Apr 2022 [cited 19 Dec 2023]. Available: https://www.mhlw.go.jp/toukei/saikin/hw/iryosd/20/dl/09gaikyo02.pdf

2. How many of these centers perform kidney biopsies?

Response: As of July 18, 2023, there are 715 teaching facilities accredited by the JSN [1]. Although an exact number of accredited facilities performing kidney biopsies is unavailable, the 2018 survey on nationwide kidney biopsy practices targeted these accredited facilities [2]. We have added this to the Participants portion of the Methods section as follows:

“As of July 18, 2023, there are 715 teaching facilities accredited by the JSN [1]. Although the exact number of facilities performing kidney biopsies is unavailable, we believe that the number is close to the number of these accredited teaching facilities.”

References

1. Japanese Society of Nephrology. Teaching facilities accredited by the Japanese Society of Nephrology (715 facilities as of July 18, 2023) (In Japanese). In: Japanese Society of Nephrology [Internet]. 18 Jul 2023 [cited 19 Dec 2023]. Available: https://jsn.or.jp/jsninfo/about/facilities/

2. Kawaguchi T, Nagasawa T, Tsuruya K, Miura K, Katsuno T, Morikawa T, et al. A nationwide survey on clinical practice patterns and bleeding complications of percutaneous native kidney biopsy in Japan. Clin Exp Nephrol. 2020;24: 389–401. doi:10.1007/s10157-020-01869-w

3. How many nephrologists certified by the Japanese Society of Nephrology (JSN) were practicing in Japan at the time of the survey? How may were invited to participate in the survey? How many responded to the invitation?

Response: There were 5,777 certified nephrologists at the time of the survey [1]. The sampling method used was convenience sampling, using the mailing list for JSN members or direct mailing by members of the working group to nephrologists of their acquaintance [2]. We have added this to the Participants portion of the Methods section. 

Because we invited respondents via convenience sampling for the survey, the total number of individuals who were invited was unfortunately unavailable. Thus, we have added the following sentence to the limitation portion of the discussion section: 

“In addition, the total number of individuals who were invited to complete the survey was unavailable.”

The number of respondents was 434, as documented in the Results section.

References

1. Japanese Medical Specialty Board. List of the numbers of members and certified physicians of each society (as of August 2021) (In Japanese). In: Overview of the Japanese medical specialty system, 2021 [Internet]. Mar 2022 [cited 19 Dec 2023]. Available: https://jmsb.or.jp/wp-content/uploads/2022/04/gaiho_2021.pdf

2. Wada T, Shimizu S, Koizumi M, Sofue T, Nishiwaki H, Sasaki S, et al. Japanese clinical practice patterns of primary nephrotic syndrome 2021: a web-based questionnaire survey of certified nephrologists. Clin Exp Nephrol. 2023. doi:10.1007/s10157-023-02366-6

4. Please provide a map of the centers, indicating the number of practicing nephrologists and the number of nephrologists who completed the survey.

Response: We have provided a Google My Map representing the working facilities of the nephrologists who responded to the survey and have named it Figure 1 ( http://tinyurl.com/4j2ct6s9 ). The number of nephrologists per facility who completed the survey was one in 77% (215 facilities) of the facilities, two in 13% (37 facilities), and three or more in 9% (26 facilities). Unfortunately, the number of nephrologists working at the facility was not available. We have added the following sentence to the Results section:

“The locations of which the ZIP codes (assuming there was only one nephrology provider in that district) were correctly identified among the 278 facilities are shown in Figure 1.”

As responded in comment #3, 434 nephrologists responded to the survey.

---

## [Decision Letter · Decision Letter 2]

5 Feb 2024

Japanese Clinical Practice Patterns of Rituximab Treatment for Minimal Change Disease in Adults 2021: A Web-Based Questionnaire Survey of Certified Nephrologists

PONE-D-23-34256R2

Dear Dr. Shimizu,

We’re pleased to inform you that your manuscript has been judged scientifically suitable for publication and will be formally accepted for publication once it meets all outstanding technical requirements.

Kind regards,

Justyna Gołębiewska

Academic Editor

PLOS ONE

Additional Editor Comments (optional):

Reviewers' comments:

Reviewer's Responses to Questions

**Comments to the Author**

1. If the authors have adequately addressed your comments raised in a previous round of review and you feel that this manuscript is now acceptable for publication, you may indicate that here to bypass the “Comments to the Author” section, enter your conflict of interest statement in the “Confidential to Editor” section, and submit your "Accept" recommendation.

Reviewer #2: All comments have been addressed

2. Is the manuscript technically sound, and do the data support the conclusions?

Reviewer #2: Yes

3. Has the statistical analysis been performed appropriately and rigorously? 

Reviewer #2: Yes

4. Have the authors made all data underlying the findings in their manuscript fully available?

Reviewer #2: Yes

5. Is the manuscript presented in an intelligible fashion and written in standard English?

Reviewer #2: Yes

6. Review Comments to the Author

Reviewer #2: I have no more comments.

7. PLOS authors have the option to publish the peer review history of their article (what does this mean?). If published, this will include your full peer review and any attached files.

Reviewer #2: No

---

## [Editor Report · Acceptance letter]

21 Mar 2024

PONE-D-23-34256R2 

PLOS ONE

Dear Dr. Shimizu, 

I'm pleased to inform you that your manuscript has been deemed suitable for publication in PLOS ONE. Congratulations! Your manuscript is now being handed over to our production team.

Kind regards, 

on behalf of

Dr. Justyna Gołębiewska 

Academic Editor

PLOS ONE